# Robotic Active Information Gathering for Spatial Field Reconstruction with Rapidly-Exploring Random Trees and Online Learning of Gaussian Processes [note 1]

**DOI:** 10.3390/s19051016

**Published:** 2019-02-27

**Authors:** Alberto Viseras, Dmitriy Shutin, Luis Merino

**Affiliations:** 1German Aerospace Centre (DLR), Oberpfaffenhofen, 82234 Weßling, Germany; dmitriy.shutin@dlr.de; 2School of Engineering, Universidad Pablo de Olavide (UPO), 41013 Seville, Spain; lmercab@upo.es

**Keywords:** robotics, information gathering, Gaussian processes (GPs), rapidly exploring random trees (RRT)

## Abstract

Information gathering (IG) algorithms aim to intelligently select a mobile sensor actions required to efficiently obtain an accurate reconstruction of a physical process, such as an occupancy map, or a magnetic field. Many recent works have proposed algorithms for IG that employ Gaussian processes (GPs) as underlying model of the process. However, most algorithms discretize the state space, which makes them computationally intractable for robotic systems with complex dynamics. Moreover, they are not suited for online information gathering tasks as they assume prior knowledge about GP parameters. This paper presents a novel approach that tackles the two aforementioned issues. Specifically, our approach includes two intertwined steps: (i) a Rapidly-Exploring Random Tree (RRT) search that allows a robot to identify unvisited locations, and to learn the GP parameters, and (ii) an RRT*-based informative path planning that guides the robot towards those locations by maximizing the information gathered while minimizing path cost. The combination of the two steps allows an online realization of the algorithm, while eliminating the need for discretization. We demonstrate that our proposed algorithm outperforms state-of-the-art both in simulations, and in a lab experiment in which a ground-based robot explores the magnetic field intensity within an indoor environment populated with obstacles.

## 1. Introduction

Information gathering (IG) is a fundamental task in a wide range of robotic applications such as environmental monitoring [1], or magnetic field intensity mapping [2]. The objective is to collect information efficiently by deciding on the actions of a robot—a mobile sensor, while optimizing the resources employed, e.g., available energy or time. This may be economically advantageous or even life-critical in search and rescue missions.

Here two key issues arise. First, an underlying model of the physical process under study is employed. By modeling spatial variations of the physical process, we can fill spatial gaps between measurements using interpolation or extrapolation [3]. The stronger the correlations and the better they are represented in a model, the fewer measurements are needed to achieve a certain reconstruction accuracy. Additionally, the use of a model together with some information metric (e.g., expected uncertainty reduction or future information gain) allows a robot to predict the impact of certain robot actions and states. Second, an active perception/sensing strategy is used to maximize the mentioned metric. These strategies can be classified as myopic (e.g., [4,5]), where the next sensing action is selected in a greedy fashion by maximizing some information metric, i.e., without accounting for future behaviour of the algorithm. Alternatively, non-myopic strategies [6,7,8] sense the process of interest and plan several steps ahead based on the previously acquired data. In this work we are considering the latter approach.

In IG with a mobile sensor, an appropriate information metric plays a crucial role, and a number of such metrics has been proposed for this purpose. Entropy [9], empowerment [10], Fisher information [11] or mutual information (MI) [12] exemplify well possible choices. The calculation of the information metric requires an underlying model for representing the observed process. In this work, we propose the use of GPs for such purpose [3]. GPs represent a powerful method to model spatial phenomena.

The use of MI for active sensing with GPs has been extensively studied by Krause et al. [13]. However, they consider two assumptions that make their approach not suitable for online IG problems. First, the authors assume that the parameters that define the GPs covariance function are a priori known, which does not typically hold. Additionally, they assume discrete sensor placements and do not consider the robot’s motion and constraints. Furthermore, the computation of MI is expensive for online IG.

Several works that relax the first assumption (i.e., the known hyperparameters) have been also proposed in the literature [2,14]. However, in Refs. [2,14] discrete sensor placements are assumed. In contrast to these works we derive an algorithm that is able to handle a complex environment, i.e., an environment that is populated with obstacles, and does not require a spatial gridding of explored space.

The central question that we address in this paper is how to efficiently gather information of an unknown physical process, which takes place in a complex environment, with a robot that runs an algorithm online as it collects information. We solve this problem by proposing a two-step strategy. First, the algorithm finds a highly informative location according to a pre-specified information metric. This location we term a *station*—a concept inspired by frontiers in autonomous robotic exploration [15]. Once the station is found, the robot updates its GP models and plans a path towards the station. When planning, we trade-off path cost with the information gained while traversing the path to the station. This we term informative path planning (IPP). Once the robot reaches the station it will update its GP model. Then the robot will look for the next station, planning a new path by leveraging the updated model.

The combination of the two steps (station search and IPP) permits an online realization of the algorithm; i.e., our proposed algorithm does not require a preprocessing step. We validate the IPP and the full IG strategy separately and highlight the performance increase respect to state-of-the-art algorithms. This validation we perform in simulations, as well as in an experiment that we carried out in our lab. In particular, we demonstrate how a robot running our proposed algorithm online outperforms traditional IG methods to map a magnetic field intensity, while assuming no prior knowledge about the magnetic field process.

Let us also point out that the work presented here is largely based on our previous publication [16], yet it extends the latter in several important respects. In particular,
We provide a more detailed description of the algorithms and the underlying methods. This helps the reader to better understand the algorithm implementation and simulation results.This paper also performs an analysis of the algorithm’s computational complexities.Also, additional simulations are presented and analyzed. Specifically, we carry out a detailed analysis of the proposed RRT*-based informative path planner. Moreover, we include an additional scenario to test the whole exploration strategy described in the paper. Moreover, a metric that benchmarks state-of-the-art algorithms according to their solution quality is also introduced.In this paper we also include an evaluation of the online learning of the GP hyperparameters and discuss the effect of online hyperparameter learning on the algorithms performance.Finally, we include an experiment with a real robot performing on exploration and reconstruction of a magnetic field using a sensor. We describe in detail the experimental setup and discuss the obtained results.

The remainder of the paper is organized as follows. We present in Section 2 the related work. Section 3 states formally the problem. In Section 4 we describe the proposed IG algorithm. Section 5 presents the analysis performed to validate the algorithm through simulations. Section 6 describes the experimental results, followed by conclusions.

## 2. Related Work

Traditional IG algorithms typically assume a discrete state space, and do not take into account robots’ dynamics [2,14,17]. Authors in Ref. [14] derive an algorithm that guides a robot towards a position that (i) “explores” the model to learn the optimal GP parameters according to the current measurements as fast as possible, and (ii) exploits the current model to gather as much information as possible. The developed algorithm uses a non-myopic approach that, however, considers neither the robot’s dynamics constraints, nor the presence of obstacles. The problem of learning the optimal hyperparameters while gathering information has been tackled as well in Ref. [2]. The authors propose a decentralized graph-based greedy multi-agent algorithm, where each of the robots gathers information about the process of interest within an obstacle-free environment, learns the optimal hyperparameters given the current measurements, and avoids inter-agent collisions. In Ref. [17] the authors propose an algorithm to gather information in a graph environment under temporal logic constraints. In contrast to traditional IG algorihtms, IPP aims to plan paths that take into account robots’s dynamics.

IPP englobes algorithms that aim to plan a path which is both feasible given the robot’s differential constraints and optimal with respect to some information metric that is calculated according to an a priori known or learned model. IPP has been proposed in the literature to solve problems such as the online localization of radio-tagged wildlife with an UAV [18], target localization and coverage [19], autonomous soaring [20], environmental mapping [21,22], ecological studies [23], weed active classification [24] or IG [25,26]. Our focus in this paper lies specifically in IPP for IG tasks.

Optimal IG with a multi-robot system was tackled in Ref. [27]. There the authors assume a grid environment that is obstacle-free. The assumption of a grid environment presents two major drawbacks. First, the introduction of the cell’s size adds an additional parameter to the algorithm. Second, it prohibits the generalization of the algorithm to robots with a large state space [26]; typically larger than four states [28]. Sampling based path planning algorithms [28,29] are natural candidates for solving such problems. In particular, we propose the adaptation of the asymptotically optimal rapidly exploring random trees (RRT*) [29] for IG tasks by the incorporation of an information metric in the algorithm.

Sampling based planning algorithms were initially proposed in the literature to solve the deterministic path planning problem. They were extended to applications such as handling of uncertainty in the robot’s pose and motion [30], robot’s tracking [11] or IG [25]; to name only a few.

The incorporation of information metrics in the RRTs has been already investigated in the literature. The information-rich RRT (iRRT) extends the RRT algorithm by incorporating a Fisher information measure [11]. However, Ref. [11] is limited to tracking applications. Also, RRTs have been used for exploration tasks with an UAV [31]. There the authors generate several alternative trajectories with the RRT algorithm and select the one that results in the highest mutual information between the current estimate and the corresponding prediction conditioned on a selected route. We will use this algorithm to benchmark the performance of our proposed IPP. Ref. [32] proposes the use of rapidly exploring random cycles (RRC) for persistent monitoring of a spatio-temporal Gaussian random field. Our focus lies, however, on efficient IG of a static physical process; i.e., we aim to gather the maximum information in the minimum amount of time.

One of the most relevant IPP works in last years is Ref. [25]. There, the authors propose the rapidly exploring information gathering (RIG) planner—a sampling based algorithm that is able to solve the IG problem under a pre-specified budget constraint. They assume that the underlying model that describes the process is a priori known, and the robot does not need to reach a particular goal position.

Our work addresses a similar problem as Ref. [25] but it differs in two principal aspects: (i) our algorithm does not require prior information of the physical process, which, in contrast to Ref. [25], allows an online realization of the algorithm; (ii) our algorithm introduces a trade-off between information gathering and a cost of a particular selected path. The consequence of last aspect leads to a path objective function that incorporates both an information metric as well as a cost term. In conjunction with intelligent station selection, the IPP represents the key contribution of this work.

## 3. Problem Statement

We wish to gather information about an a priori unknown physical process with a robot as accurately as possible, in the sense of minimizing the Root Mean Squared Error (RMSE) between a process estimate (given by a GP model) and the (unknown) ground truth. Our goal is to perform an exploration that is efficient given the available resources. To this end, we devise in this paper movement strategies so as to reduce the model’s uncertainty over the exploration space as efficiently as possible. The reduction of the model’s uncertainty is performed in this work through the minimization of the process entropy. This has been shown to be effective to reduce the RMSE between a process estimate and the (unknown) ground truth [33,34].

To achieve this, we make a few simplifying assumptions. Specifically, we assume the following:The physical process takes place in an environment populated with obstacles. The borders and obstacles that define the environment are a priori known. This assumption allows us to abstract the exploration of the physical process from the perception and mapping of the environment.The physical process is time-invariant during the information gathering task.The robot’s position is known exactly and is noise-free. We assume that there exists an external positioning system that provides us with a highly accurate localization, e.g., a Real Time Kinematic Global Positioning System (GPS-RTK) for outdoor scenarios, or a motion tracking system for indoor cases. Uncertainty in positioning can also be accounted for using e.g., GPs [35].

The robot position will be denoted by $x_{r}\in X_{free}\subset R^{n_{d}}$, where $X_{free}$ corresponds to the free space in the robot’s configuration space, and $n_{d}$ is the dimensionality of the robot’s state space. The physical process at position $x\in X_{free}$ is given by the variable y(x)∈R. Typically, however, the process value is not observed directly, but measured using some sensor. Here we assume a simple sensor model that represents a measured process as z=y(x)+ϵ(x), where *z* is a scalar measurement taken by the robot at position x and ϵ(x) is a random noise that models the sensor’s noise. In the following we will assume that, for different measurements, noise samples ϵ are independent and identically distributed according to $\epsilon \left(x\right)∼N\left(0,\sigma_{n}^{2}\right)$, i.e., they follow a Gaussian distribution with zero mean and variance $\sigma_{n}^{2}$. Notice that we do not include the dependency of x in *z* to simplify notation.

In this work we propose an algorithm that allows a robot to autonomously decide where to measure next in order to reconstruct accurately and efficiently y(x) at any position $x\in X_{free}$. Naturally, this requires a model that can accurately represent the observed phenomenon. Here we make use of GPs for this purpose. In the following we give a short outline of GPs for modelling spatial data.

## 4. Gaussian Processes for Spatial Data

A GP is a collection of random variables, any finite number of which have a joint multivariate Gaussian distribution [3]. It is fully specified by a mean function m(x) and a covariance function $k(x,x^{\prime },\theta )$ for any given positions x, $x^{\prime }$ and some hyperparameters θ. In this work we assume that m(x) is set to zero, which implies an absence of a priori known values of the observed process. As covariance function, we employ a squared exponential (SE) [3] because of its ability to model smooth processes. The SE is defined by $k\left(x,x^{\prime },\theta \right)=\sigma_{f}^{2}\exp\left(-\frac{||x-x^{\prime }{\left|\right|}^{2}}{2l^{2}}\right)+\sigma_{n}^{2}\delta_{xx^{\prime }}$, where $\delta_{xx^{\prime }}=1$ iff $x=x^{\prime }$ is the Kronecker’s delta, and $\theta ={\left[\sigma_{f}^{2},l,\sigma_{n}^{2}\right]}^{T}$. *l* is the so-called characteristic length-scale (informally, “how close” two points x and $x^{\prime }$ have to be to influence each other significantly); $\sigma_{f}^{2}$ represents the maximum allowable covariance; and $\sigma_{n}^{2}$ is the variance of the noise fluctuations [3].

Now, let us make the following definitions: $X={\left[x^{\left[1\right]},x^{\left[2\right]},\cdots ,x^{\left[n\right]}\right]}^{T}$ is a matrix where each row correspond to a spatial location where a robot has taken a measurement, $z={\left[z^{\left[1\right]},z^{\left[2\right]},\cdots ,z^{\left[n\right]}\right]}^{T}$ are the corresponding measurements, and $X_{*}={\left[x_{*}^{\left[1\right]},x_{*}^{\left[2\right]},\cdots ,x_{*}^{\left[p\right]}\right]}^{T}$ is a matrix where rows correspond to “probe” locations—points in space where we predict the process value using the learned model. Furthermore, using $k(x,x^{\prime },\theta )$ we define $K,K_{*},K_{**}$ as follows: (1)$$K=\left[\begin{array}{ccc} k(x^{\left[1\right]},x^{\left[1\right]}) & \cdots & k(x^{\left[1\right]},x^{\left[n\right]})\\ \vdots & \ddots & \vdots \\ k(x^{\left[n\right]},x^{\left[1\right]}) & \cdots & k(x^{\left[m\right]},x^{\left[n\right]}) \end{array}\right],\;K_{*}=\left[\begin{array}{ccc} k(x^{\left[1\right]},x_{*}^{\left[1\right]}) & \cdots & k(x^{\left[1\right]},x_{*}^{\left[p\right]})\\ \vdots & \ddots & \vdots \\ k(x^{\left[n\right]},x_{*}^{\left[1\right]}) & \cdots & k(x^{\left[n\right]},x_{*}^{\left[p\right]}) \end{array}\right],\;K_{**}=\left[\begin{array}{ccc} k(x_{*}^{\left[1\right]},x_{*}^{\left[1\right]}) & \cdots & k(x_{*}^{\left[1\right]},x_{*}^{\left[p\right]})\\ \vdots & \ddots & \vdots \\ k(x_{*}^{\left[p\right]},x_{*}^{\left[1\right]}) & \cdots & k(x_{*}^{\left[p\right]},x_{*}^{\left[p\right]}) \end{array}\right].$$

Let us stress that k(·) and subsequently K, $K_{*}$, and $K_{**}$ are all functions of θ. Notice that we do not include this dependency to simplify notation.

Given z and X, we can predict the process values $y_{*}$ and the corresponding uncertainties at $X_{*}$. The vector $y_{*}$ is a random vector with the following probability density function: $p(y_{*}|X_{*},X,z)=N$$\left(\mu_{*},\Sigma_{*}\right)$, where $\mu_{*}$, $\Sigma_{*}$ are calculated as (see Ref. [3] for more details):(2)$$\begin{array}{cc} \mu_{*} & =m\left(X_{*}\right)+K_{*}^{T}K^{-1}\left(z-m\left(X\right)\right),\\ \Sigma_{*} & =K_{**}-K_{*}^{T}K^{-1}K_{*}. \end{array}$$

Learning a GP model implies an estimation of θ from process observations z. Given the training data X and z, we can estimate the hyperparameters $\theta_{*}$ that best fit our measurements; this is also known as model training. The hyperparameter estimation is typically done by maximizing the log-marginal likelihood (LML) with respect to θ,
(3)$$\theta_{*}=\underset{\theta }{\mathrm{argmax}}\left\{-\frac{1}{2}z^{T}K^{-1}z-\frac{1}{2}\log\left|K\right|\right\}.$$

This is a nonlinear optimization problem that requires application of numerical optimization techniques [3].

## 5. Efficient Information Gathering Using RRT-Based Planners and GPs

### 5.1. Algorithm Overview

We aim to explore with a robot an unknown process y(x), for $x\in X_{free}$. As we employ GPs as underlying model of y(x), the lack of prior information about y(x) implies that hyperparameters θ need to be estimated and updated as the robot collect measurements. In fact, the spatial distribution of the information metric is directly related to the values of θ. An adaptation of θ while the robot moves will essentially make any planning suboptimal, since the information metric computed at any region in space will follow the fluctuations of the hyperparameter estimates $\theta_{*}$. Instead, we propose updating θ only at some point in the vicinity of the robot’s current location that maximizes the information gained about the modeled process. This point we name it a *station*, a concept inspired by frontiers in autonomous robotic exploration [15]. Then, given $\theta_{*}$, the robot plans a route towards the station so as to further increase the amount of information about y(x). In this case the resulting information metric calculated at all points in space is fixed and thus planning (conditioned on $\theta_{*}$) will optimize the desired utility function.

A block diagram of the whole scheme is shown in Figure 1. We present in Algorithm 1 a detailed pseudo-code. Our proposed algorithm works as follows. First, the robot learns the $\theta_{*}$ that best model the previously acquired measurements z, X (line 3 in Algorithm 1). This is done with Equation (Equation 3).
**Algorithm 1.**$\mathrm{SBSRE}\;\mathrm{Algorithm}(x_{r},X_{free},b,StopAlgorithm)$
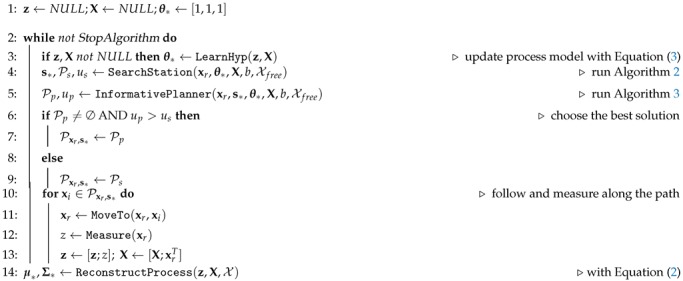


Once the robot estimates $\theta_{*}$, it searches for a highly informative station $s_{*}$ (line 4) using Algorithm 2. Algorithm 2 takes as an input $x_{r}$ and $X_{free}$, a budget constraint on the path cost *b*, and $\theta_{*}$ that allow the robot to calculate the expected information contained at a station. In addition to $s_{*}$, the algorithm outputs a suboptimal, yet feasible path $P_{s}=\left[x_{r},∆,x_{i},∆,s_{*}\right]$, with $x_{i}\in X_{free}$, and the corresponding path utility $u_{s}$. More on the computation of the utility and its properties will be discussed in Section 5.3.

Then the robot plans a trajectory from $x_{r}$ to $s_{*}$ (line 5) using an informative path planner (Algorithm 3) in order to refine $P_{s}$. In Section 5.3 we describe Algorithm 3 in more details. The algorithm result is a trajectory $P_{p}$, together with its corresponding utility $u_{p}$ that trades off the information gathering with the path’s cost. Algorithm 3 has an anytime nature; i.e., it aims to find a feasible solution and then improves it with time. Please note, however, that it is possible that for the stop criterion pre-defined by the user, e.g., planning time, Algorithm 3 is either not able to find a path or the found path is of worse quality (in terms of used utility) than $P_{s}$. To guarantee that a solution is found, the robot compares solutions from Algorithms 2 and 3 in lines 6–9 of Algorithm 1, and select the best path $P_{x_{r},s_{*}}$ according to the information metric.

Finally, the robot follows $P_{x_{r},s_{*}}$ until it reaches the station $s_{*}$, while taking new measurements $y(x_{i})$, with $x_{i}\in P_{x_{r},s_{*}}$, and updating accordingly vector z and matrix X to its knowledge database (lines 10–13, Algorithm 1). Then, the main loop is repeated until some stopping criterion is fulfilled, e.g., maximum exploration time, or the remaining process uncertainty. Once the robot finishes gathering information, it can predict the value of the process $\mu_{*}$ and the associated uncertainty $\Sigma_{*}$ of the prediction for any $X\subset X_{free}$ using Equation (Equation 2) (line 14, Algorithm 1).
**Algorithm 2.**$\mathrm{SearchStation}(x_{r},\theta_{*},X,b,X_{free})$
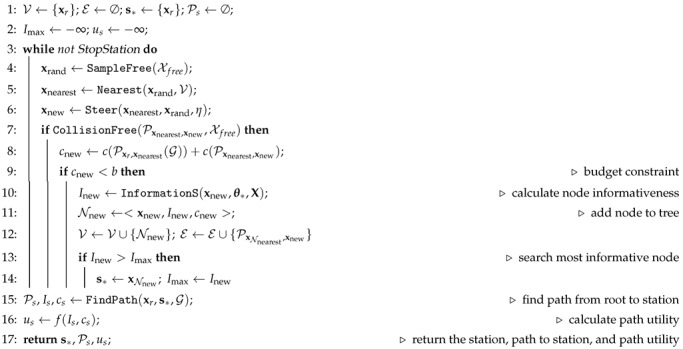


### 5.2. Search for Highly Informative Stations

Let us now consider the searching algorithm for a new station in more detail. A station is a location $s^{\prime }\in X_{free}$ that is highly informative according to a pre-specified information metric (see Figure 2). In addition, the search of a station must fulfill the following two requirements: (i) it must be reachable for the robot; (ii) its calculation must have an anytime nature to allow the online realization of the algorithm. To realize these requirements, we propose an adaptation of the kinodynamic Rapidly-exploring Random Trees (RRT) algorithm [28] where we extend the RRT nodes to incorporate an information measure. The RRT algorithm has an anytime nature, and fulfills the first requirement since it is able to account for the robot’s kinematics and avoid possible collisions with obstacles. Please note that in SearchStation we are not concerned about the optimality of the path, but rather about reachability of the station. Therefore we employ a simple, yet suboptimal path planner such as RRT, which provides a quick way to “sort out” stations that are not reachable by the robot. Using e.g., RRT* [29] for realizing this test is possible, but computationally less efficient.

Formally, the search of $s_{*}$ can be formulated as:(4)$$s_{*}=\underset{s^{\prime }\in X_{free}}{\mathrm{argmax}}I\left(s^{\prime }\right)\;s.t.\;c\left(P_{x_{r},s^{\prime }}\right)\leq b,$$
where $I(s^{\prime })$ is a measure of the expected information at $s^{\prime }$ (the particular measure employed is described in Section 5.4 later), $c(P_{x,s^{\prime }})$ is the cost of traversing $P_{x,s^{\prime }}$, and *b* is a trajectory budget. We assume $c(P_{x,x^{\prime }})$ as strictly positive, additive (If we have two partial trajectories $P_{1,2}$ and $P_{2,3}$ that can be concatenated to yield a trajectory $P_{1,3}$, a cost function is considered additive if $c\left(P_{1,3}\right)=c\left(P_{1,2}\right)+c\left(P_{2,3}\right)$) and monotonically increasing. Examples of such cost measures are the total time or fuel required to traverse the path; the number of measurements taken along the path can also serve as valid cost function. Here we choose time as path cost. Please note that *b* is employed to limit the search space per single planning step. This essentially implies that paths that exceed a cost budget are pruned from the RRT.

The key steps of the SearchStation algorithm are summarized in Algorithm 2. Algorithm 2 is an extension of RRT for IG tasks. The first modification with respect to RRT consists of an extension of the standard RRT node N∈V. Like in Ref. [25], in Algorithm 2 a node includes (i) the spatial location of node $x_{\mathrm{new}}$, (ii) the expected information $I_{\mathrm{new}}$ at $x_{\mathrm{new}}$, and (iii) the cost $c_{\mathrm{new}}$ of reaching $x_{\mathrm{new}}$ from $x_{r}$ (line 11). The latter is computed using the robot motion model, while $I_{\mathrm{new}}$ is calculated with the function InformationS (line 10). For more details on the information calculation we refer the reader to Section 5.4. In addition to RRT we include a budget constraint *b* (line 9) as well.

The ultimate goal of Algorithm 2 is selecting $s_{*}$ that has the highest information (lines 13–14). In addition we calculate the path $P_{s}$ that drives the robot from its current position to the station with function FindPath (line 15). This function also outputs the information $I_{s}$ and the total cost $c_{s}$ of $P_{s}$. Given $I_{s}$ and $c_{s}$ we can calculate the utility $u_{s}$ of $P_{s}$ (line 16). Details about the calculation of utility function (Equation 6) and information of the path (Equation 7) are given in Section 5.3 and Section 5.4, respectively.

### 5.3. Informative Path Planner Using RRT*

The goal of the informative path planner is to refine $P_{s}$ calculated with Algorithm 2. Here we aim to calculate a path that fulfills the following two requirements: (i) it is feasible given the robot’s dynamics and does not incur collisions with obstacles; and (ii) it is efficient, in the sense of maximizing the information gathering, while minimizing the path cost. Formally, we aim to find the optimal path $P_{x_{r},s_{*}}$ between states $x_{r}$ and $s_{*}$. This can be formulated as the following optimization problem:(5)$$\begin{array}{cc} \underset{P_{x_{r},s_{*}}\subset X_{free}}{\mathrm{argmax}} & f(I\left(P_{x_{r},s_{*}}\right),c\left(P_{x_{r},s_{*}}\right))\\ s.t.: & c(P_{x_{r},s_{*}})<b \end{array}$$

Here I(·) and c(·) are the functions that evaluate the information and cost of the path, respectively, f(·,·) is a function that evaluates the information-cost trade-off (the utility), and *b* is a budget for the path cost. We summarize the InformativePlanner in Algorithm 3.

Algorithm 3 is an extension of RRT* [29] for IG tasks. Algorithm 3 allows a robot to gather information efficiently and autonomously by exploiting the current GP model, defined by $\theta_{*}$ and learned online as the robot gathers measurements (line 3, Algorithm 1). In contrast to RRT*, here we replace the concept of the path cost by the concept of utility. The utility *u* of a path is a value that weights the importance of a path. In this paper, we formulate the utility so that it fulfills our IG objective. That is, we aim to gather as much information as possible along the path towards $s_{*}$ while generating trajectories with the minimum cost. This implies that $f(I\left(P_{x_{r},s_{*}}\right),c\left(P_{x_{r},s_{*}}\right))$ should grow with $I(P_{x_{r},s_{*}})$ and decrease as $c(P_{x_{r},s_{*}})$ becomes large. Like in Refs. [36,37,38], we represent this trade-off with the following function:
(6)$$f\left(I\left(P_{x_{r},s_{*}}\right),c\left(P_{x_{r},s_{*}}\right)\right)=\alpha \frac{I(P_{x_{r},s_{*}})}{c(P_{x_{r},s_{*}})}.$$
with α a coefficient balancing the trade-off between I(·) and c(·). $c(P_{x_{r},s_{*}})$ is chosen as a time needed to traverse $P_{x_{r},s_{*}}$. $I(P_{x_{r},s_{*}})$, which we calculate with the function InformationP, will be explained in detail in Section 5.4. Let us also emphasize that Equation (Equation 6) allows us to extend the algorithm to applications where taking a measurement is expensive. In contrast to prior work, our algorithm incorporates a information-cost trade-off that permits accounting for measurements that incur a high cost.

#### Non-Monotonicity of the Utility Function

The definition of Equation (Equation 6) introduces an additional difference between RRT* and Algorithm 3, since Equation (Equation 6) is non-monotonic. The non-monotonicity of Equation (Equation 6) compromises the optimality guaranty of RRT* [29]. Despite this, our simulation results suggest that Algorithm 3 is still able to approach the optimal solution, as shown in Section 6.2. Furthermore, the non-monotonicity of Equation (Equation 6) requires the inclusion of a mechanism to avoid the creation of cycles in the tree. A cycle is a sequence of vertices starting and ending at the same vertex [39]. Cycles might appear in the graph due to our specific choice of the utility function, which is not guaranteed to increase monotonically with the growing tree. Cycles can be created during the rewiring process if a node $N_{\mathrm{near}}^{\prime }$, which belongs to the path that connects the robot’s position with $N_{\mathrm{new}}$, could be reached with a higher utility from $N_{\mathrm{new}}$ than its previous utility. This problem does not arise with a monotonic utility function, since the inclusion of a new node always incurs a higher cost. Here, however, a longer path could have a higher utility if we gather more information along it. To generalize RRT* to a larger class of utility functions, such as Equation (Equation 6) or the one used by Ref. [36], we propose a procedure to eliminate cycles. This is implemented in function CyclesFree (line 22). This function takes as input $x_{\mathrm{new}}$ and $V_{\mathrm{near}}$. Then it iterates over $N_{\mathrm{near}}\in V_{\mathrm{near}}$, and removes those nodes $N_{\mathrm{near}}=\left\{x_{\mathrm{near}},I_{\mathrm{near}},c_{\mathrm{near}}\right\}$ where $x_{\mathrm{near}}\in P_{x_{r},x_{\mathrm{new}}}\left(G\right)$.

Once the robot finishes the execution of the algorithm, which is given by the StopPlanner criterion, it calculates the best path $P_{x_{r},s_{*}}$ in terms of utility with function FindBestPath. This function connects $s_{*}$ to x∈V∈G that are closer than a distance η from it. Then we calculate the utility of all those possible paths and choose the one with the highest utility. The utility, together with the computed path form the algorithm output. In case the algorithm does not find a suitable path, it outputs an empty path with minus infinite utility.

### 5.4. Information Metric

Algorithm 1 relies on an information metric to evaluate the informativeness both of a $x\in X_{free}$ and of a $P_{x,x^{\prime }}\subset X_{free}$. In particular, we argue about the use of Mutual Information (MI) [40] and mean entropy as information metrics. To finalize, we motivate our particular choice: mean entropy as information metric.

#### 5.4.1. Mutual Information

MI has been extensively employed in the IG literature [12,13,25]. Indeed, it seems to be a perfect fit for selecting informative sampling locations because it takes into account the cross-correlations of the test points. However, we observed that MI is not adequate for algorithms that require an extensive computation of the information metric, as it is also pointed out in Ref. [34]. Due to the long computation time of the MI for GPs [40], most of the planning time would be dedicated to calculating MI instead of adding nodes to the tree in Algorithms 2 and 3. This reduces the size of the explored region of the state space given a certain exploration time, resulting in a loss of performance. There exists efficient algorithms to calculate MI, like e.g., the one introduced in Ref. [41]. However, to the best of our knowledge, they are not applicable to GPs.
**Algorithm 3.**$\mathrm{InformativePlanner}(x_{r},s_{*},\theta_{*},X,b,X_{free})$
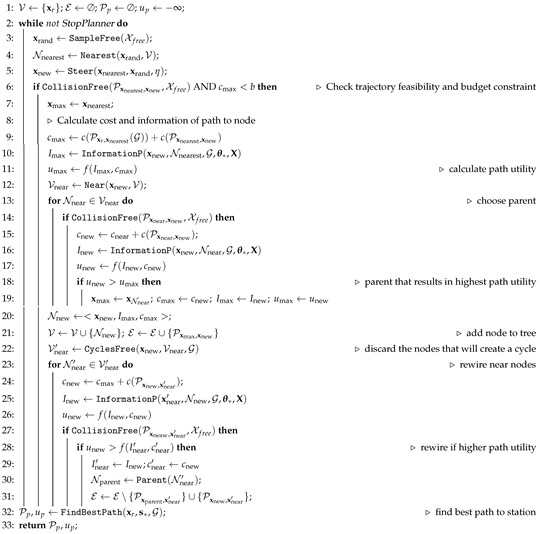


#### 5.4.2. Mean Entropy

As a viable alternative, we decided to use mean entropy $\bar{H}\left(P_{x,x^{\prime }}\right)$ as information metric. This is primarily motivated by the concept of entropy rate, which is a limit of the joint entropy as the number of observations grows [40]. Mean entropy converges to entropy rate as a special case. Formally we define $\bar{H}\left(P_{x,x^{\prime }}\right)$ as:(7)$$I\left(P_{x,x^{\prime }}\right)=\bar{H}\left(P_{x,x^{\prime }}\right)=\frac{1}{|P_{x,x^{\prime }}|}\sum_{x_{i}\in P_{x,x^{\prime }}}H(x_{i}),$$
where $H(x_{i})$ is the entropy at $x_{i}$. The latter can be easily calculated for GPs [3].

$\bar{H}\left(P_{x,x^{\prime }}\right)$ presents a diminishing property [13] similar to the MI. For example, imagine two paths that have the same sum of entropies. An averaged entropy would favor the one that requires fewer measurements. This is a desirable property that would be crucial if we considered the cost of taking a measurement. Note also that $H(x_{i})$ only needs to be calculated once for each $x_{i}\in V$, as we can save $H(x_{i})$ and reuse it each time we evaluate a path that includes $x_{i}$. In summary, $\bar{H}\left(P_{x,x^{\prime }}\right)$ is computationally efficient and favors those paths that have higher information at smaller cost.

In the following, we specify the calculation of functions InformationP, InformationS, and FindPath. InformationP computes the information of $P_{x_{r},x_{\mathrm{new}}}$ with Equation (Equation 7). In contrast InformationS computes the information of a single $x_{\mathrm{new}}\in X_{free}$—a potential $s_{*}$. This is also calculated with Equation (Equation 7), where $P_{x,x^{\prime }}=\left[x_{\mathrm{new}}\right]$. Finally, FindPath searches $P_{x_{r},s_{*}}$ with highest utility.

### 5.5. Computational Complexity

In this section we perform an assessment of the algorithm’s overall time complexity. To this end we split the analysis in two main parts: updating the GPs model, which corresponds to estimating θ, and planning $P_{x_{r},s_{*}}$.

The complexity of the estimation of θ is given by the inversion of the K matrix, given by Equation (Equation 3). The basic complexity of this matrix inversion is $O(n^{3})$, with *n* the number of collected measurements [3].

The planning of $P_{x_{r},s_{*}}$ corresponds to the execution of Algorithms 2 and 3, which are based on RRT and RRT*, respectively. For $N_{p}$ tree’s nodes, the basic complexity of RRT and RRT* is $O(N_{p}\log N_{p})$ to create the tree, and $O(N_{p})$ to query the best path from the tree. We refer to [28,29] for a detailed analysis of the algorithms’ time complexity.

In addition to the basic RRT and RRT* algorithms we must include the computation of the information metric (Equation 7), which has a time complexity $O(n^{3})$ [3]. As we explained in Section 5.2 and Section 5.3 the calculation of the information metric is delimited to positions $\left\{x\in X_{free}\;|\;||x-x_{r}{\left|\right|}_{2}^{2}\leq b\right\}$ by path cost budget *b*. The presence of a trajectory budget allows us to reduce the time complexity of the information metric calculation from $O(n^{3})$, which is unbounded and grows with the number of measurements *n*, to $O(n_{b}^{3})$ with $n_{b}$ the number of measurements that lie within the region defined by path cost budget *b*. Please note that $n_{b}<<n$.

In summary, the computational complexity of the overall algorithm (one iteration of Algorithm 1) is $O(N_{p}\log N_{p}+n_{b}^{3})$, with $n_{b}<<n,N_{p}$. We remark that RRT and RRT* algorithms have an anytime nature. Therefore, $N_{p}$ will depend on the running time of the planning algorithm, which is a user-defined criterion.

## 6. Simulations and Discussion of Results

In this section we present the simulations setup and performance results of Algorithm 1. We divide this analysis in two main parts. First, we compare Algorithm 3 against two state-of-the-art algorithms in an IG task of a physical process with a priori known parameters. Second, we evaluate Algorithm 1 in a more complex IG task of an unknown physical process that takes place within an environment populated with obstacles.

### 6.1. Simulations Setup

Here we describe the simulation setup used to validate Algorithm 1. We employ data that we store as grids of 20×20 cells. In particular, we use synthetic data with a resolution of 5 cm in Section 6.2, and real data with a resolution of 10 cm in Section 6.3. For the simulation we assume a round-shaped ground-based holonomic toy robot with 5 cm radius that moves with a constant speed of 0.2 m per second. We also consider a two-dimensional state space ($n_{d}=2$). Here we employ a holonomic-robot to abstract the active sensing strategy from the robot’s motion. This is a common strategy employed in works like e.g., [25], one of our benchmark algorithms. However, note that Algorithm 1 is valid for a n-dimensional state space, and for arbitrary robot’s dynamics given the restrictions imposed by RRT or RRT* algorithms. We also assume that the robot needs an infinitesimally small time to take a measurement. The robot can move in a continuous space, and we assume that measurements taken within one cell of the grid are equal.

For the SearchStation and InformativePlanner algorithms we select the following parameters: the parameter η in the Steer function and the distance employed by the function Near to search neighbors nodes are both set to the measurement’s resolution, i.e., 5 or 10 cm depending on the concrete simulation. We select the running time as stop criterion for the SearchStation and InformativePlanner algorithms, with a value of 5 and 10 s respectively. We consider a trajectory budget b=10 s, which corresponds to a planning horizon of 2 m given the robot’s speed. We initialize θ to the following values l=1, $\sigma_{f}=1$, $\sigma_{n}=0.1$. We use the pyGPS library [42] to perform the GPs regression and learning of θ. We carry out each of the simulations 40 times.

### 6.2. Analysis of the Informative Path Planner

#### 6.2.1. Setup

First, we analyze the individual performance of Algorithm 3. This assumes an available GP model with fixed hyperparameters. The goal of Algorithm 3 is, given this model, to find the trajectory that optimizes (Equation 5) as fast as possible. Since our information metric corresponds to the mean entropy (Equation 7), we simulate three scenarios with distinct entropy structures (see Figure 3):Scenario 1 recreates a physical process with low spatial correlation in which a robot has already gathered two patches of measurements. The blue areas correspond to the measured areas and the red areas to the non-measured positions. We employ the following θ: l=0.02, $\sigma_{f}=0.084$, $\sigma_{n}=0.02$.Scenario 2 recreates the same scenario, but now we consider a process with higher spatial correlation. Here we set l=0.13, $\sigma_{f}=0.084$, $\sigma_{n}=0.09$.Scenario 3 corresponds to three measurements that are taken randomly for each of the simulation runs. For this case we consider the same hyperparameters as for scenario 2.

We fix $x_{r}$ to (x=0.2,y=0.5) and $s_{*}$ to (x=0.8,y=0.5) for all the simulations runs.

#### 6.2.2. Choice of the Information Function

In this section we compare the performance of the informative path planner using our proposed information metric Equation (Equation 7) and maximizing MI. We carry out the analysis for scenario 2, as it is the one that presents higher spatial correlation, which translates into a higher informativeness of Equation (Equation 7). We compare Equation (Equation 7) with MI, in terms of: (i) time to find a first $P_{x_{r},s_{*}}$ ($t_{first}$); (ii) posterior differential entropy that results after measuring along $P_{x_{r},s_{*}}$ output by Algorithm 3; and (iii) cost of $P_{x_{r},s_{*}}$. Please note that the random variable associated to the process after measuring along $P_{x_{r},s_{*}}$ is a GP, which is a continuous random variable. Therefore, we refer to differential entropy, instead of entropy that is defined for discrete random variables. Let us remark that differential entropy can take negative values.

We show the results of the evaluation in Table 1. According to Table 1, Equation (Equation 7) finds a first path seven times faster than MI, reduces the posterior differential entropy by one half, and the path calculated with Equation (Equation 7) has a slightly smaller cost than the one calculated with MI. This lets us conclude that our proposed information metric, given by Equation (Equation 7), outperforms the MI in an online sensing setting that requires an extensive computation of the information metric.

#### 6.2.3. Performance Analysis

We benchmark Algorithm 1 against two state-of-the-art sampling-based informative path planning algorithms:(i) the technique of [31], where multiple paths are obtained by running the RRT planner several times, and the paths are then evaluated according to the information metric. This algorithm we will term MultiplesRRT;(ii) the RIG-tree planner [25], to which we will refer as RIGAlgorithm.

In both cases, motivated by results from Table 1, we employ (Equation 6). For the RIG-tree we use one of the approaches suggested by the authors in Ref. [25]. Specifically, we consider the pruning based on the heuristic that the utility function is modular. Additionally, we defined two nodes as co-located if they are within the same cell of the grid. For more details about the implementation, we refer the reader to the original paper [25]. We also tested the other alternatives proposed by the authors, but they offered a lower performance in our particular setup.

Moreover, the two benchmark algorithms are not designed to reach a particular goal. This makes a comparison with our algorithm difficult. We solve this by selecting all samples that are closer than a distance η from goal $s_{*}$, and then connecting them to $s_{*}$. This results in paths that link $x_{r}$ with $s_{*}$. We analyze in Figure 4 the performance of compared algorithms as a function of planning time.

**Utility analysis.** The difference in terms of utility (see first row of Figure 4) with respect to the other algorithms ranges between 0.05 and 0.15 bits per second. We notice as well that the RIG algorithm only presents a minor improvement of the utility as the planning time increases. We believe this is due to the inclusion of the goal constraint, which the RIG algorithm is not able to handle.

**Algorithm complexity analysis.** Another important figure that characterizes the algorithm is the number of nodes spanned by the path planner. We observe in the second row of Figure 4 that the MultiplesRRT variant has a limited number of nodes since we reset the algorithm each time we find a new path. Furthermore, Algorithm 3 requires a larger number of nodes than the RIGAlgorithm. The latter employs a smaller number of nodes because of the pruning strategy that removes those co-located nodes that have a smaller utility than the new added node. However, this does not lead to a higher complexity per iteration, as we can observe in the third row of Figure 4, which shows the number of iterations of the algorithm vs. the planning time. Here, the MultiplesRRT alternative offers the lowest complexity.

**Paths output by Algorithm 3.** In the last row of Figure 4 we depict the paths output by Algorithm 3. We observe that, for scenario 1, the robot takes the path that has the most information and takes the least time, which results in a straight line. However, in scenario 2 the straight line corresponds to a path that has little information, and therefore the robot takes a path that is longer but allows it to gather more information as it visits not yet measured locations. These results illustrate the need for defining a utility function that trades off the information gathering and the path’s cost (Equation 6).

**Posterior entropy analysis.** We showed in Figure 4 that Algorithm 3 outperforms the considered state-of-the-art approaches in terms of our information function. However, this does not necessarily imply that our algorithm can find a more informative path. To make a fair comparison between the three considered algorithms we evaluate them in terms of the posterior entropy after measuring along the calculated path. In addition, we compare the cost of the resulting paths. Table 2 shows the results for scenario 2, for 180 s of planning time. We can conclude that Algorithm 3 offers the best ratio entropy-cost for all scenarios. Additionally Algorithm 3 results in a twofold and sevenfold increase respect to RIGAlgorithm and MultiplesRRT, respectively, while offering a similar path cost.

### 6.3. Analysis of the Exploration Strategy

#### 6.3.1. Setup

We validate in this section the proposed exploration approach, as summarized in Algorithm 1, in an environment populated with obstacles. To this end we carried out simulations in two environments that present different obstacles structure (see Figure 5). Scenario A emulates a corridor with different rooms, while scenario B considers thicker block-like obstacles. Simulations employ real data, collected with a ground-based robot at Deutsches zentrum für Luft- und Raumfahrt (German Aerospace Center) (DLR), which corresponds to a magnetic field intensity in an indoor environment [2].

#### 6.3.2. Performance Analysis

As we previously saw, informative path planner (Algorithm 3) is superior to MultiplesRRT and RIGAlgorithm. Algorithm 3 goal is, given an a priori known model, to calculate an informative path between robot’s current location and a goal position. However, in an IG task we typically have no knowledge about the process model. Therefore we proposed in this paper Algorithm 1, which is able to explore an a priori unknown process.

In the literature, two common approaches to deal with an exploration of an unknown process are a myopic approach, and a random trajectory. Therefore, we used these two to benchmark Algorithm 1:Myopic approach: the next station is selected from one of robot’s neighboring cells, with the cell size given by the ground truth data (see Figure 5). The robot selects the cell that has the highest entropy as measured by Equation (Equation 7) [1,2,14].Random approach: an RRT is grown from $x_{r}$ for the same planning time and budget *b* as in SearchStation algorithm. The next station is selected randomly as one of the leaves of the RRT. The path that links $x_{r}$ to the selected leaf is then followed by the robot.

**RMSE analysis.**Figure 6 shows the mean and variance of the RMSE for all executions. This is done for the different strategies and for both scenarios. We also test the methods under assumption that the optimal θ are known and fixed (listed with an asterisk sign). Our goal is to shift the Mean(RMSE) curve to the left bottom corner. This implies a small RMSE that is achieved efficiently in terms of time resources. First fact that we can observe is that Algorithm 1 is able to obtain a performance comparable to Algorithm 1*, which uses pre-learned hyperparameters. This result indicates that Algorithm 1 performs a correct exploration-exploitation trade-off. That is, in the beginning of the IG task, we would expect a robot to perform exploratory actions to learn about the environment and about process of interest. Once this is done, the robot can use this knowledge to update and exploit the GPs model. In contrast, note that myopic and random strategies cannot perform a correct exploration-exploitation trade-off, which is reflected in their inferior performance when hyperparameters are unknown.

Additionally, another interesting finding is that a random strategy outperforms a myopic one. This is due to an exploration-exploitation trade-off. A random strategy performs purely exploration, which implies that it does not use the previously acquired data to plan next path. This has the advantage that is able to explore much more space than the myopic strategy. In contrast, the myopic strategy does pure exploitation as it uses the learned GP model to select the next best possible position. In this case, for the two scenarios considered, we can see that exploration is more important than exploitation. In fact, Algorithm 1 balances between exploration and exploitation. This results in a higher performance of Algorithm 1 respect to the other strategies. The myopic approach with optimal θ is the only one that offers a comparable performance after the 900 s mission. Please note, however, that it assumes an a priori known model, which is unrealistic for an actual IG task.

**Solution quality.** We analyze in Figure 7 the quality of the solution respect to the best possible performance that we could obtain by systematic sampling. We consider the best possible solution as the estimation over the complete environment that results after measuring at all $x\in X_{free}$. This we term it $RMSE_{best}$. Let us remark that this solution considers optimal θ. More formally we define solution quality as: $Solution\;quality=\frac{1}{n_{sim}}\sum_{i=1}^{n_{sim}}100\frac{RMSE_{best}}{RMSE_{i}}$. We show in Figure 7 the solution quality for a myopic, random, and Algorithm 1. A percentage of 100% indicates that Algorithm 1 is able to achieve an RMSE that is equal to the best possible RMSE that we could obtain. According to Figure 7, after 900 s Algorithm 1 is able to obtain a RMSE that is the 90%, while the myopic and random approach achieve only half of it.

**Comparison with RIG algorithm.** To get a better understanding of Algorithm 1 capabilities we include a comparison with state-of-the-art RIGAlgorithm. Specifically, we consider the following for the RIGAlgorithm: (i) the model is a priori known; i.e., we know the GPs hyperparameters and they do not need no be estimated, (ii) the utility function corresponds to the MI, as suggested by the authors in Ref. [25], and (iii) the planning time is 600 s and then we let the robot follow and measure along the planned path. Let us remark that these are favorable conditions for the RIGAlgorithm as our algorithm assumes an a priori unknown model that needs to be estimated online. Please also note that complexity of RIGAlgorithm is $O(N_{p}^{3})$. Algorithm 1 complexity is $O(N_{p}\log N_{p}+n_{b}^{3})$, with $n_{b}<<n,N_{p}$. We run the simulation 40 times starting from different positions in the environment. Then we calculated the RMSE after measuring along the calculated path. The average RMSE that we obtained for the RIGAlgorithm is 0.27, which is much higher than the one obtained by Algorithm 1 that is 0.05 (see Table 3). We believe that the lower performance of the RIGAlgorithm is due to the fact that the algorithm grows a single tree to explore the complete environment. Notice that the complexity of adding a new sample grows exponentially as the tree grows, which complicates the exploration of the complete environment. In contrast, Algorithm 1 runs multiple consecutive trees using our devised two-step approach that permits an efficient online exploration.

#### 6.3.3. Hyperparameters Analysis

Finally, in Figure 8 we show the evolution of the estimated θ (mean and variances) for Scenario A. Results correspond to the the average values calculated over 40 simulation runs. To estimate θ we optimize the Log-Marginal-Likelihood (LML) (Equation 3). The LML is a differentiable function and, therefore, conjugate gradients are a proper alternative to obtain $\theta_{*}$ [3]. Let us remark that the non-convexity of the LML could drive the optimizer to local minima. To overcome this issue we run the optimization algorithm several times (10 to be specific) with initial values drawn randomly from a uniform distribution defined over the set of feasible hyperparameter values. Then we pick the best solution.

Figure 8 allows us to demonstrate that Algorithm 1 permits a robot to learn the hyperparameters θ online. This is in fact crucial, as incorrect hyperparameters result in a model mismatch that would lead the robot to take incorrect actions. To obtain a better understanding of the hyperparameters learning, we compared Algorithm 1 against a myopic and a random approach. In this case we can observe that θ values for Algorithm 1 converge slightly slower than the other approaches. In the myopic and random approach, the process is re-estimated more often as compared to Algorithm 1. This can explain a faster convergence of θ. However, a slower convergence of θ does not imply an inferior performance in terms of the RMSE respect to these two strategies, as shown in Figure 6 and Figure 7. We can also observe that $\sigma_{n}$ for the random trajectory converges to a slightly higher value compared to the myopic approach and Algorithm 1. We believe this is due to the fact that the random trajectory often repeats measurements at the same positions and this has an impact on the learned θ.

## 7. Experiments and Discussion of Results

Finally we test the algorithm in an experiment employing a real ground-based holonomic robot that is used to autonomously explore a magnetic field intensity within an indoor laboratory environment populated with obstacles (see Figure 9). A video (see Appendix A) that shows the real-time execution of the experiment is attached as an online resource, and can be visualized in https://youtu.be/lV9ntxRmvr4.

### 7.1. Experimental Setup

The IG task takes place in an environment that measures 3 by 6 m. It contains 8 boxes of different sizes that are arbitrary placed. Prior to the experiment we scan the magnetic field intensity with a resolution of 10 cm, which yields a 1800 cells grid. The magnetic field intensity ranges between 5 and 84 μT (see Figure 9). These measurements are then used as ground truth to test the performance of Algorithm 1. Note that considering these measurements as ground truth is a realistic assumption since the magnetometer can be assumed to be noise-free according to its specifications.

We use the algorithm parameters described in Section 6.1. We run Algorithm 1 in a central computer and then we send the corresponding waypoints to the robot using the Robot Operating System (ROS) [43]. The central computer is equipped with an Intel Core i7-6600U CPU, and 8 GB of physical memory. The robot is equipped with a Raspberry Pi 2 model B that runs the robot’s controller to guide the robot to the desired position.

### 7.2. Experimental Results

We benchmark the performance of Algorithm 1 against a greedy and a random trajectory. We show in Figure 10 screenshots of the algorithm execution for three instants of time. Specifically, we show the estimation of the process (left), the entropy of the process model (middle), and the tree that was produced by the robot to plan a path toward the next station (right). The robot is represented with a circle and the path planned by Algorithm 3 is depicted as a thick red line. We can draw two main conclusions. First: the robot covered all areas except the ones that are occupied by obstacles, as exemplified in Figure 10 (central column) where blue areas correspond to explored positions. Second: the process estimation at 946 s, which corresponds to the end of the IG task, is almost identical (despite the boxes) to the ground truth data shown in Figure 9.

Finally, Figure 11 illustrates the evolution of the RMSE for the three compared strategies. We remark that the RMSE is computed over all positions of the environment that are reachable for the robot. Additionally, we would like to remark that Algorithm 1 does not require any preprocessing steps, as GPs hyperparameters are computed online during the IG tasks using Equation (Equation 3). As we concluded in Section 6, here we verify that Algorithm 1 greatly outperforms the myopic and random trajectories. Our proposed strategy is able to decrease the RMSE to approx. 1.24μT, which represents approx. a fifteen-fold improvement respect to the other approaches.

## 8. Conclusions and Future Work

In this work we presented a strategy that profits from the active sensing approach for efficient information gathering in complex environments with a robotic mobile sensor whose motion can be planned with an RRT-like algorithm [28]. We assume that a process of interest can be represented with GPs. The GP model allows the robot to capture spatial correlations of the process by adapting (i.e., training) online the GP hyperparameters to the measured sensors data. The trained model will then be used by the robot to make predictions about the process value and the corresponding uncertainty at locations not yet visited. In fact, we employ the predicted process uncertainty to guide a robot trajectory: it moves to locations—termed stations in this work—where the uncertainty (entropy) within a certain range from the current position is the highest. We realize the search of stations with a modification of the RRT algorithm. This allows us to perform an efficient search in space of highly informative locations while, at the same time, allows the robot to trade off the exploration-exploitation with respect to the GP hyperparameters. For realistic environments, however, there exist multiple trajectories that can be used to reach a station from the current position of the robot; also, not all of the trajectories are feasible due to, e.g., presence of obstacles. This requires a certain path planing mechanism. To solve a path planing problem we use the RRT* algorithm. In contrast to classical RRT* approaches, here we proposed a modification of the RRT* algorithm that also exploits the learned GP model for more efficient planning. Specifically, the algorithm trades-off the cost of the generated path with an information gained (measured as a mean entropy along a trajectory) while traversing the generated path. In this way, the trajectory toward the station is generated so as to maximize the information gain along the shortest possible path. In addition, we included a mechanism (removal of graph cycles) that allows us to generalize RRT* to non-monotonic utility functions.

We validated the proposed algorithm in simulations using real data, as well as in an experiment with a robot that explores a magnetic field intensity in a laboratory populated with obstacles. Results show that the devised informative path planner offers a twofold and sevenfold increase, in terms of the posterior entropy of the process after the robot measures along the planned path, with respect to [25,31], respectively. Moreover, the IG strategy proposed in this work achieves a fifteen-fold improvement respect to a random walk and an entropy-driven myopic approach. Additionally, we showed that our method that does not require prior knowledge of the process of interest in order to perform an efficient exploration as our algorithm learns model parameters online while exploring.

As future work, we aim to extend the proposed algorithm to handle information gathering tasks that are closer to real-world applications. This would include accounting for more complex robots, as well as uncertainty both in the robot’s pose and motion. In addition we should consider the mapping of the environment. The inclusion of these extensions could be accounted for by defining a cost function that trades-off these elements. For example, this would imply guiding the robot to a position where it gathers not only information but it will also help it to get a better understanding of the environment in which it operates. Another natural extension for the algorithm is to consider the multi-robot case. This way we could benefit from the correct coordination between robots to achieve a much higher performance. This would require coping with challenges such as: (i) what information should the robots transmit to achieve a correct coordination, (ii) how to avoid inter-agent collisions, (iii) how to achieve appropriate consensus to maximize the global information gain in an efficient manner. We are currently working to solve some of the aforementioned issues. In addition, our goal is to validate our approach in field experiments both with ground-based robots and a swarm of micro aerial vehicles (MAVs). 

## Figures and Tables

**Figure 1 sensors-19-01016-f001:**
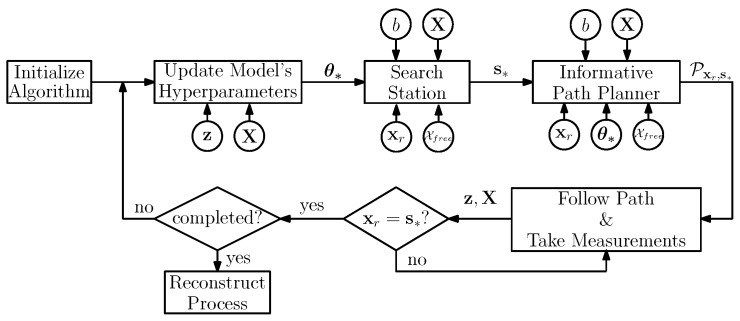
Algorithm block diagram.

**Figure 2 sensors-19-01016-f002:**
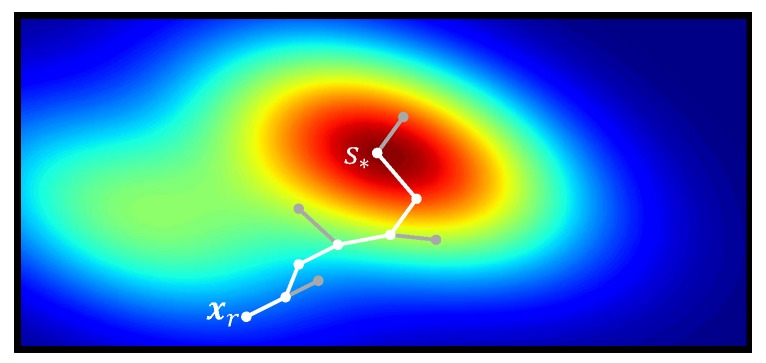
Search for highly informative stations. The color scale represents the informativeness, as measured by a predefined information metric, at a particular location. In particular, dark blue corresponds to low informativeness and red represents high informativeness. Algorithm 2 selects $s_{*}$ as the location with the highest informativeness among all x∈V.

**Figure 3 sensors-19-01016-f003:**
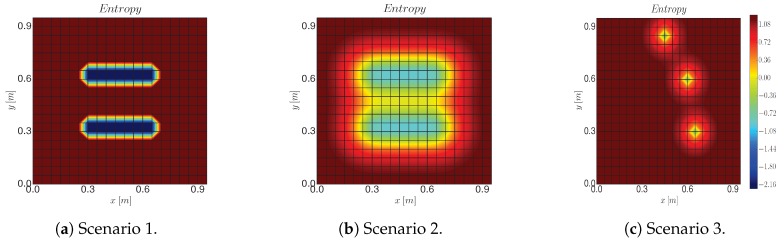
Simulation scenarios used to test the performance of Algorithm 3. We represent the entropy of the process after measuring at some spots. Scenario 1 and 2 correspond to two patches of measurements of processes that have a low and high spatial correlation, respectively. Scenario 3 corresponds to three measurements taken at random positions for each simulation run. In this latter case, we show just one example.

**Figure 4 sensors-19-01016-f004:**
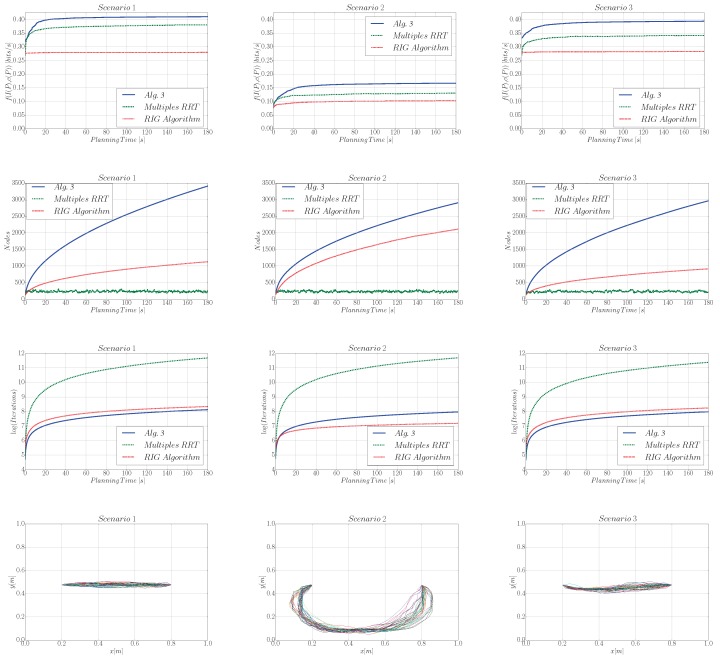
Performance analysis of Algorithm 3 as we increase the planning time. Here, from the first to the last row, we evaluate the utility of the best path, the number of nodes spanned by the tree, and the algorithm’s complexity that is represented as the curve number of iterations vs. planning time. In addition, we plot the 40 paths output by Algorithm 3 given 180 s of planning time.

**Figure 5 sensors-19-01016-f005:**
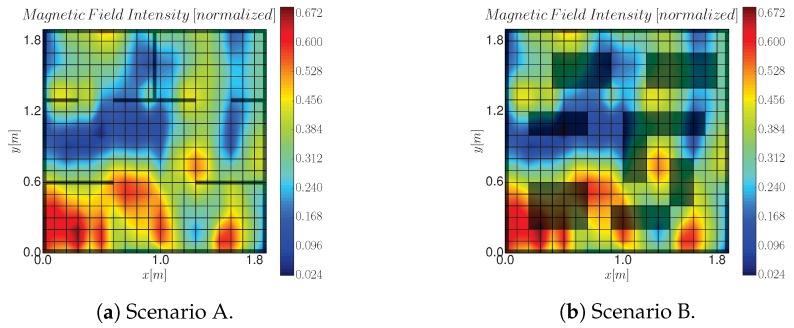
Scenarios employed to test Algorithm 1. Black polygons correspond to the obstacles and the underlying picture is the magnetic field intensity we aim to explore.

**Figure 6 sensors-19-01016-f006:**
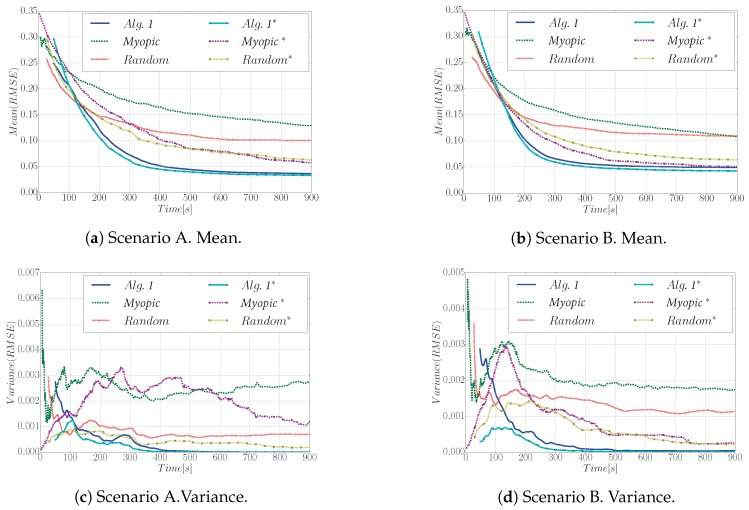
RMSE between the estimation of the process and the ground truth. Top: scenario A. Bottom: scenario B. We represent the mean and variance of the RMSE over the 40 simulations we carried out. Here we test three different trajectories: (i) Algorithm 1, (ii) a myopic approach, and (iii) random trajectories. For all of them we compare their performance assuming: (i) no prior knowledge about the process, which implies an online learning of θ, and (ii) assuming they know the optimal θ a priori (marked with an asterisk).

**Figure 7 sensors-19-01016-f007:**
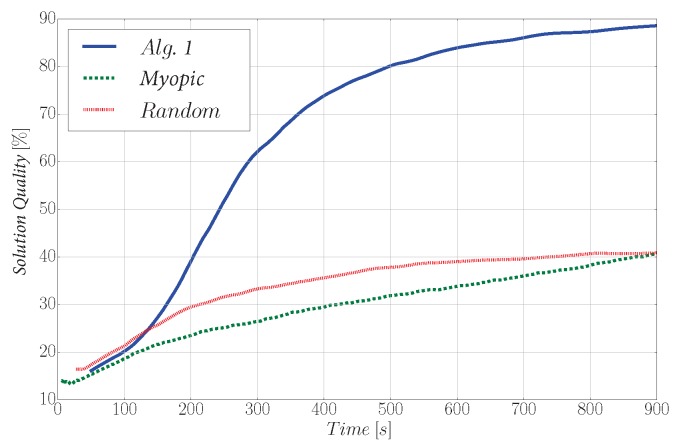
Quality of the solution achieved by Algorithm 1, a myopic, and a random trajectories after a 900 s mission. A 100% corresponds to the best possible solution. These curves correspond to the mean value achieved over the simulations carried out for scenarios A and B.

**Figure 8 sensors-19-01016-f008:**
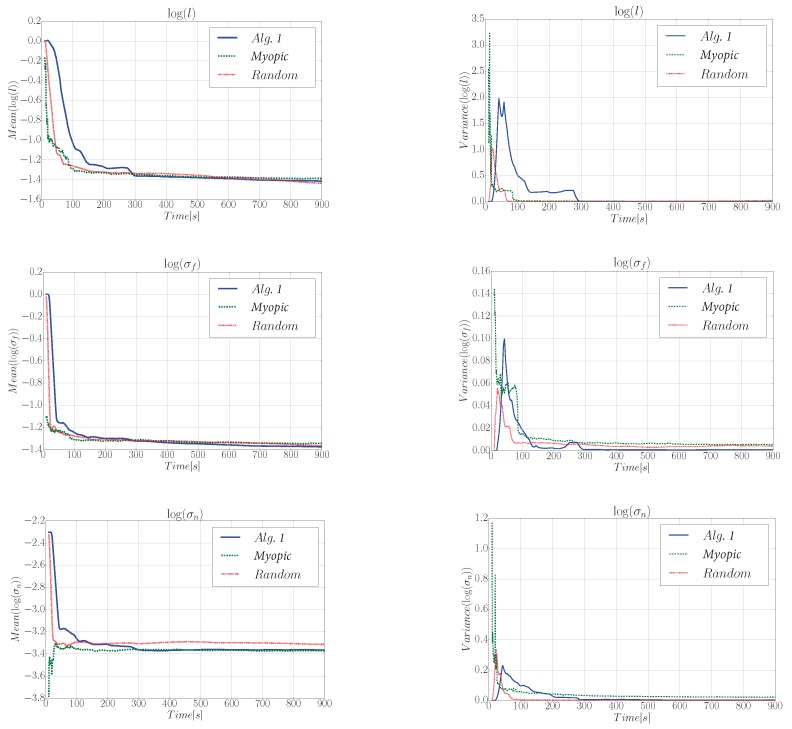
GPs hyperparameters learned during the information gathering task for Algorithm 1, a myopic, and a random trajectory. We represent the mean and variance over the 40 simulation runs. We show the hyperparameters $\theta ={\left[\sigma_{f}^{2},l,\sigma_{n}^{2}\right]}^{T}$.

**Figure 9 sensors-19-01016-f009:**
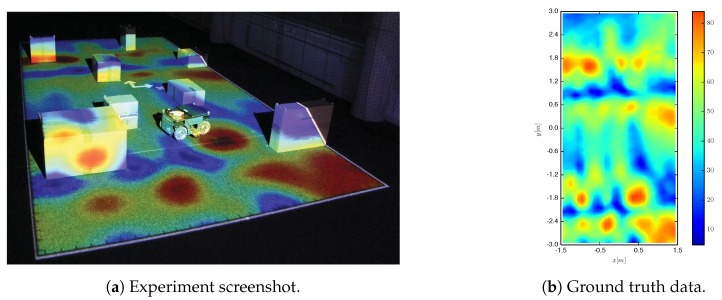
Ground-based robot exploring the magnetic field intensity within an indoor environment populated with obstacles. The projection on the ground corresponds to the actual magnetic field intensity, which we measured prior to the experiment to use it as ground truth. The magnetic field intensity ranges between 5 and 84 μT.

**Figure 10 sensors-19-01016-f010:**
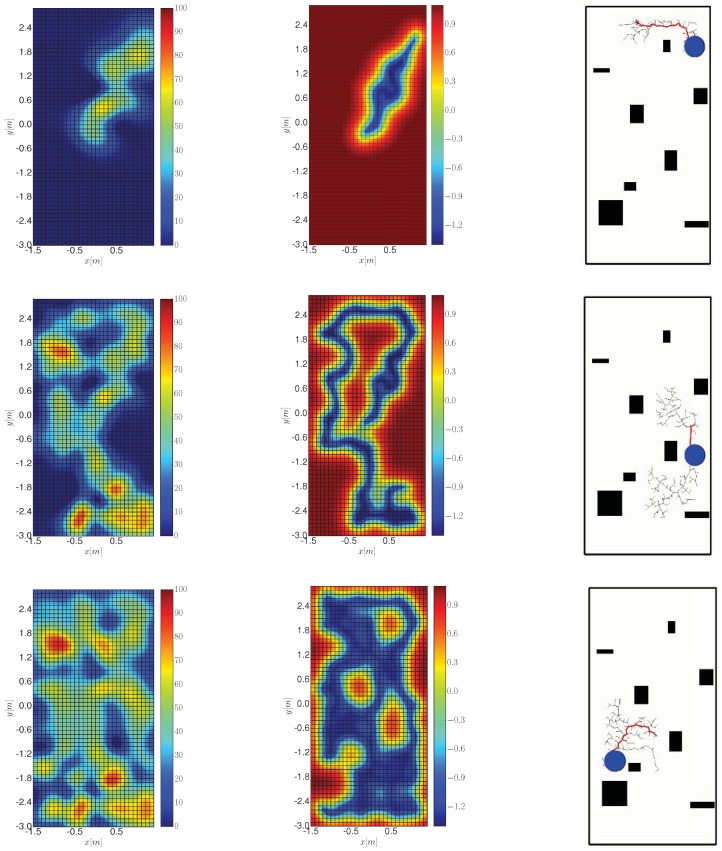
Screenshots of Algorithm 1 execution. The three rows correspond to three instants of times: 133, 502 and 946 s. From left to right, the columns are the estimation of the process (measured in μT), the entropy, and the planned path using Algorithm 3.

**Figure 11 sensors-19-01016-f011:**
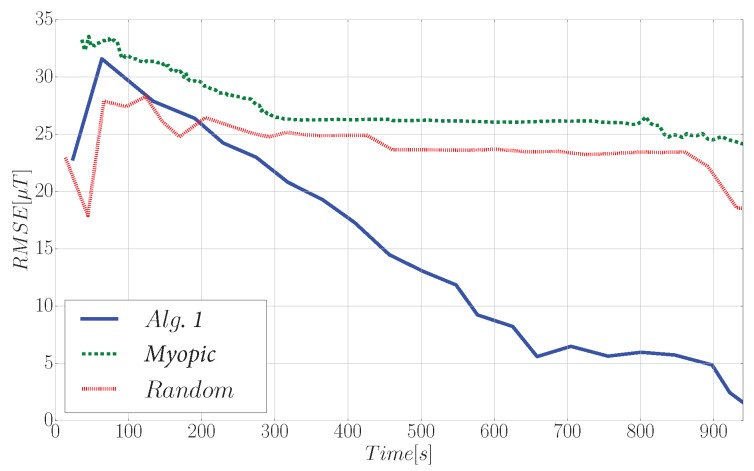
Evolution of the RMSE during a 940 s exploration task that was carried out with a ground-based robot in our lab.

**Table 1 sensors-19-01016-t001:** Analysis of the information function. We compare Equation (Equation 7) with the MI, in terms of: time to find a first $P_{x_{r},s_{*}}$ ($t_{first}$), process differential entropy that results after measuring along $P_{x_{r},s_{*}}$ output by Algorithm 3, and cost of the resulting path. Results correspond to the average ±3σ variation, calculated over 40 simulations runs.

	$t_{\mathrm{first}}$ [s]	Differential Entropy [bits]	Cost [s]
Mean Entropy (Equation 6)	6.31±2.1	−6.93±0.18	6.79±0.06
Mutual Information	46.71±3.2	−3.54±0.12	6.86±0.08

**Table 2 sensors-19-01016-t002:** Posterior differential entropy and path cost evaluated over the completeenvironment after measuring $P_{x_{r},s_{*}}$, calculated for 180 s of planning time.

	Differential Entropy [bits]	Path Cost [s]
Algorithm 3	−6.93±0.18	6.79±0.06
MultiplesRRT	−3.68±0.32	6.23±0.09
RIGAlgorithm	0.03±0.29	5.65±0.12

**Table 3 sensors-19-01016-t003:** RMSE at t=600s resulting after exploring scenario B. For this comparison Algorithm 1 employs Equations (Equation 6) and (Equation 7), and RIGAlgorithm uses MI [25].

	RMSE at *t* = 600 s
Algorithm 1	$0.05\;\pm \;10^{-5}$
RIGAlgorithm	$0.27\;\pm \;10^{-4}$

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
