# Peer review of "Robotic Active Information Gathering for Spatial Field Reconstruction with Rapidly-Exploring Random Trees and Online Learning of Gaussian Processes"

_sensors, 2019, doi:10.3390/s19051016_

Round 1

Reviewer 1 Report

Well done.

Author Response

Well done.

We would like to thank the reviewer for taking the time to read the paper. We highly appreciate the positive review.

Reviewer 2 Report

The approach is related to the concept of empowerment (D. Polani) such that it is necessary to  discuss this in paper, in particular with respect to computational complexity, which is obviously much higher for empowerment (as seen in the end of your Sect. 5.5) For the present paper, please discuss also estimates for the complexity in 3D.

Fig. 1 suggest to update the hyperparameters based on initialisation, although the text states this is done based on data. Where do z and X come from? This needs to be explained.

Although RRT is a commonly used abbreviation, it should no be used in the title and should be explained when used in the abstract.

Some measures e.g. "quality of solution" need to be made explicit in order to support reproducibility.

"We provide a more detailed description of the algorithms and the underlying methods." The paper needs to be concise and self-contained, whether this is achieved by more or less detail is not important. Also the rest of this list may be appropriate for a letter to the editor, but not for the paper. Instead, state here, what new results are presented.

Algs. 1 and 2: use "not" instead of ""!"

Fig. 3: It would be good to modify Scenario 3 to get  a more interesting path. E.g. if the two lower obstacles are closer together, would the parameter alpha be critical for the path taken, i.e. whether in between the lower two obstacles or between the upper two (if these are now further apart, but also further away)?

Fig. 8: It is not clear whether the differences are significant. Also, it appears that random or myopic may be faster. You will need to include statistics to make your point more obvious. In 477 it is merely stated that there is not necessarily an implication, but it is still not clear why the convergence of the hyperparameters is slower and whether this *can* cause problems.

I'm not sure whether I understand the advantage of not having a model (as in RIG or empowerment) and the need of establishing a full covariance matrix.

I also don't understand the assumption on the homogeneous and additive  noise (157), how can there be any differences in the information?

Eq. 3 is a bit odd, because the argument theta was dropped before from k, and k is never specified nor exemplified, e.g. how exactly does "l" (169) enter k?

These might be easily explainable, but it order to make the paper accessible to the reader, it should be improved by adding basic explanations andmore explicit fomulas.

Check spelling of "kynodinamic"

Author Response

We would like to thank the reviewer for taking the time to read the paper. We highly appreciate the detailed comments. We are sure that these will contribute to greatly improve the paper.

Next we respond to each of the reviewer comments. Additionally, we highlighted in blue the modified text in the paper.

The approach is related to the concept of empowerment (D. Polani) such that it is necessary to  discuss this in paper, in particular with respect to computational complexity, which is obviously much higher for empowerment (as seen in the end of your Sect. 5.5) For the present paper, please discuss also estimates for the complexity in 3D.

We would like to thank the reviewer for the reference suggestion. Our approach is indeed strongly related to empowerment in the sense that both approaches employ information metric to guide robot’s actions. However, the main advantage of our algorithm with respect to empowerment is that we use a GP model that describes the physical process of interest. By exploiting the GP model a robot can predict what the best possible action might be given a current estimate of the model parameters. Additionally, the model allows the robot to perform a regression over positions not being measured or vene not accessible by the robot . These two reasons why many of state-of-the-art methods employ GPs for information gathering, as stated in Sec. 1. We included empowerment as a reference of one of the possible information metrics in Sec. 1.

Regarding the computational complexity of empowerment we cannot derive a worst case complexity for empowerment, as it completely depends on the sensors and actuators at hand. Nevertheless we can state that complexity of empowerment scales exponentially with the number of sensor and actuator states, as information metric must be calculated for each pair sensor-actuator as indicated in eq. (3) in the empowerment paper.

Now, let us discuss address the computation complexity of our algorithm in 3D. It is important to remark that our formulation  is general for an n-dimensional environment. In fact, GPs can be easily generalized to N-D case. The increase in numerical complexity is only seen in more complex evaluation of kernels. Therefore, we restrict our model to a  two-dimensional environment in our experiments for the sake of simplicity. We added a remark to Sec 6.1 to make this point clear to the reader.

Fig. 1 suggest to update the hyperparameters based on initialisation, although the text states this is done based on data. Where do z and X come from? This needs to be explained.

Thank you very much for remarking this issue. We fully agree that the submitted version of the text was misleading.

In fact, hyperparameters are learned based on data. Nevertheless, for the first iteration of the algorithm this is not possible as the robot has not gathered any data yet. Therefore, we initialize the hyperparameters to (l=1, sigma_f=1, sigma_n=1) for the first iteration. For the subsequent iterations, hyperparameters are learned online based on data.

To make this clear, we modified Alg.1 to highlight that hyperparameters are learned only if z, X are not NULL.

Although RRT is a commonly used abbreviation, it should not be used in the title and should be explained when used in the abstract.

Thank you for pointing this out. We modified the title and abstract accordingly.

Some measures e.g. "quality of solution" need to be made explicit in order to support reproducibility.

Thank you for this comment. We definitely agree that an explicit expression for the “quality of solution” must be included in the text. We added the corresponding paragraph  in Sec. 6.3.2. in the new version of the manuscript.

"We provide a more detailed description of the algorithms and the underlying methods." The paper needs to be concise and self-contained, whether this is achieved by more or less detail is not important. Also the rest of this list may be appropriate for a letter to the editor, but not for the paper. Instead, state here, what new results are presented.

We would like to thank the reviewer for the comment. We modified the contributions section accordingly to highlight the new results presented in the paper.

Algs. 1 and 2: use "not" instead of ""!"

We modified it in the text for Algs. 1,2,3.

Fig. 3: It would be good to modify Scenario 3 to get  a more interesting path. E.g. if the two lower obstacles are closer together, would the parameter alpha be critical for the path taken, i.e. whether in between the lower two obstacles or between the upper two (if these are now further apart, but also further away)?

Thanks you for the comment. We agree that the location of the “measurements spots” would lead to different paths, and performance of the algorithm might be affected.

This is the reason why, for each simulation run, we simulated multiple locations of the “measurements spots” for scenario 3. This was indicated in the text (section 6.2.1), but not in the figure.

This was indeed confusing. Therefore, we included an explanation of the scenarios in Fig.3 caption.

Fig. 8: It is not clear whether the differences are significant. Also, it appears that random or myopic may be faster. You will need to include statistics to make your point more obvious. In 477 it is merely stated that there is not necessarily an implication, but it is still not clear why the convergence of the hyperparameters is slower and whether this *can* cause problems.

The main goal of the results in Fig. 8 is to demonstrate that Alg. 1 permits a robot to learn the hyperparameters online. This is in fact a crucial point and one of the key contributions of this paper. Note that without optimal hyperparameters there will be a larger  model mismatch as GP will not optimally represent the measured process. Moreover, predictions pf the entropies and process values will be biased and the  robot will not able to take correct actions. To this end we show the mean estimation of the hyperparameters, and some statistics (the variation of the estimation). By looking at the blue curve (Alg. 1) in Fig. 8 we can confirm that. Additionally, we can also observe by looking at Fig. 6, 7 that the hyperparameters that were learned contribute to a superior performance of Alg. 1 respect to the considered benchmarks.

These experiments were sufficient for us to proof the proposed concept. Nevertheless, we were also interested in comparing the learning of the hyperparameters with  respect to a random and a myopic approach, as these are the two most widely used information gathering approaches. Here we discovered that we converged to very similar hyperparameters, which is what we expected. Additionally, we also found out that learning in  Alg. 1 converges slightly slower that the other approaches. In the myopic and random approach, the process is re-estimated more often as compared to Alg. 1. This can explain a faster convergence of θ. However, a slower convergence of θ does not imply an inferior performance in terms of the RMSE respect to these two strategies, as shown in Figs. 6 and 7.

We modified the text in Sec. 6.3.3 to make it clearer.

I'm not sure whether I understand the advantage of not having a model (as in RIG or empowerment) and the need of establishing a full covariance matrix.

. The use of a model is a crucial point in our algorithm, and therefore it is our goal that is clearly understood.

Essentially, the model (a GP in our paper) provides prior information about the physical process of interest. In particular, a GP provides information about the spatial correlation of the process; i.e. how process values are correlated to each other. The importance of using a GP model is two-fold: First, by modeling spatial variations of the physical process, we can fill spatial gaps between measurements using interpolation or extrapolation. The stronger the correlations and the better they are represented in a model, the fewer measurements are needed to achieve a certain reconstruction accuracy. Additionally, the use of a model together with some information metric (e.g. expected uncertainty reduction or future information gain) allows a robot to predict the impact of certain robot actions and states. These two reasons are the main justification of why most of state-of-the-art methods employ GPs for information gathering of spatially distributed processes, as stated in Sec. 1.  The use of the covariance matrix is required to both learn the model parameters, and to perform regression. We included in Sec. 1 some text to justify the importance of using a model for information gathering.

Last but not least, we would also like to clarify that RIG uses a GP as underlying model.

I also don't understand the assumption on the homogeneous and additive  noise (157), how can there be any differences in the information?

Thank you very much for the comment. By reading the paper again we agree that the role of the noise is not 100% clear.

Epsilon(x) models the sensor’s noise. Typically, a robot’s sensor (like the magnetic field sensor in our paper) cannot sense the actual value of the process. Instead a robot’s sensor provides a value that corresponds to the actual value subject to some noise. Here we chose to model this sensor’s noise as additive and Gaussian, since this is the standard practice in GPs-based information gathering (Rasmussen, 2005).

We clarified this in Sec. 3.

Eq. 3 is a bit odd, because the argument theta was dropped before from k, and k is never specified nor exemplified, e.g. how exactly does "l" (169) enter k?

These might be easily explainable, but it order to make the paper accessible to the reader, it should be improved by adding basic explanations andmore explicit fomulas.

Thanks for this suggestion. We agree with reviewers that in the submitted version of the paper it is hard to see the dependency of hyperparameters in the K matrix, as mentioned by reviewers.

To ease the understanding, we added in Sec. 4 the explicit formula of the squared exponential covariance function. This way, the influence of hyperparameters (like e.g. “l”) in k() becomes clear.

Check spelling of "kynodinamic"

We proof-read the text, and corrected the typo.

Reviewer 3 Report

1.       First appeared abbreviated words such as RRT- should be explained;

2.       Sec 1.1 can be reorganized instead of a series of sentences;

3.       The parameter or function in equations such as Eq.(3) should be indicated;

4.       Why the mean entropy function gets the negative value instead of positive ones in Table 1;

5.       The computational hardware performance should be indicated in Sec 7.

Author Response

We would like to thank the reviewer for taking the time to read the paper. We highly appreciate the detailed comments. We are sure that these will contribute to greatly improve the paper.

Next we respond to each of the reviewer comments. Additionally, we highlighted in blue the modified text in the paper.

First appeared abbreviated words such as RRT- should be explained;

Thank you for pointing this out. We modified the title and abstract accordingly.

Sec 1.1 can be reorganized instead of a series of sentences;

Thanks for this comment. Following the reviewer’s suggestion we reorganized the contributions section, which is not any more a separate subsection.

The parameter or function in equations such as Eq.(3) should be indicated;

Thank you very much for pointing this out. We modified Sec. 4 accordingly. In particular, we wrote the explicit formula of the squared exponential covariance function. This way, the dependency of hyperparameters in equations (1-3) becomes clear.

Why the mean entropy function gets the negative value instead of positive ones in Table 1;

Thank you very much for this insightful comment, as we also agree that this property of entropy deserves a further clarification.

In fact, the random variable that represents the process of interest at points in path P_(x_r,s_*) is a Gaussian process (GP). A GP is a continuous random variable. As such, we talk about differential entropy instead of entropy like in the case of discrete random variables. The  differential entropy, however, can take negative values.

We added this explanation in Sec. 6.2.2, and substituted in the text “entropy” by “differential entropy” to make the text clear and consistent.

The computational hardware performance should be indicated in Sec 7.

Thank you for point out these missing details. The setup of our central computer is the following: Intel Core i7-6600U CPU, and 8 GB of physical memory. The Raspberry Pi model is 2 B. We added these details to Sec. 7.1.

Reviewer 4 Report

--------------A brief summary (one short paragraph) outlining the aim of the paper and its main contributions.

The paper presents a method for information gathering of a physical process that can be performed online. It uses a two step strategy. The first step finds a location that provides high information metrics based on estimations obtained with a gaussian process. The second one builds an exploration path based on the knowledge of the environment spatial distribution and the possible information gained while traversing the path. The method assumes the space and the robot position are known, and the process to be measured is time-invariant. Additionally, it is assumed noise in measurements follows a Gaussian distribution. Experiments mapping the magnetic field intensity in several virtual environments are performed, as well as a real experiment conducted with a ground holonomic robot.

-------------Broad comments highlighting areas of strength and weakness. 

The document describes how it can be distinguished among works in the literature related to information gathering and also presents the extensions to the work previously published by the authors.

This paper provides extensive explanations of the core concepts and proposed methods through detailed algorithms. It shows different evaluation metrics for each of the experiments performed. Every section is well written and in general provides plenty of analysis for each of the parts of the method.

The organisation general of the paper makes it difficult to follow, mainly in sections V and VI (that appear as 5 and 6).

Myopic and Random approaches are not well explained and at a first look, it results counter intuitive that a random strategy performs better than a myopic one.

A main issue is found in Fig 6 results. Alg. 1 and Alg. 1* show a very similar behaviour after a few tens of seconds. Can this be interpreted as Alg. 1 has reached almost all the information about the process as Alg. 1*? This is difficult to see as there is no information of the percentage of exploration of the map. It can be assumed that a considerable proportion of the environment has been explored and Alg 1 can estimate information gain as good as Alg. 1*. 

If this is true, can the estimating of the parameters of the gaussian process be stopped in order to speed up the method?

This paper describes as differences with a previous authors' work the following:

 • We provide a more detailed description of the algorithms and the underlying methods.

 • We include an analysis of the algorithm’s computational complexity.

 • We present an in-depth evaluation of the algorithm with additional simulations. Specifically, we carry out a detailed analysis of the proposed RRT*-based informative path planner. Moreover, we include an additional scenario to test the whole exploration strategy described in the paper.

 • We evaluate the online learning of the GPs hyperparameters.

 • We propose a metric that benchmarks state-of-the-art algorithms according to their solution quality.

 • We include an experiment on the exploration and reconstruction of a magnetic field using a ground robot equipped with a magnetic field sensor. We describe in detail the experimental setup and results.

After reading the previous work it can be found only a new simulation, consisting in adding a new scenario. This scenario is very similar to the reported before.

The evaluation of the online learning of the GPs hyperparameters doesn't provides new findings about the method.

The metric proposed was already presented in the previous work.

The new material is basically the experiment performed with the real robot.

-------------Specific comments referring to line numbers, tables or figures. 

In 6.2.2 the authors compare mutual information versus mean entropy. It is mentioned in other parts of the document that 40 simulations were performed but it is not clear if that was the case for this experiment and in that case it would be necessary to show the degree of variation.

The paper states the method can be performed online, however the authors don't clarify what are the arguments and data that supports this affirmation.

It seems that the budget constrain is restricted to only one path and can not be defined in a general way as explained in [23].

Fig. 11 shows RMSE evolution in time using Random, Myopic and Alg. 1 strategies. The proposed strategy (Alg. 1) seems to obtain the lowest value but there is no explanation why a random strategy is better than a myopic one. These results repeat in Fig. 7 where the quality of the solution looks better for a random strategy.

Lines 535-538 state that the proposed path planner offers an increase over state of the art algorithms but it doesn't specify in which metric.

A more illustrative example of estimated and ground truth of the process is required as the comparison between figures 9 and 10 (as mentioned in 502-505) is difficult to obtain.

In [23] it is mentioned that the computational requirement for the RIG method is given by O(N+f(N)) where f(N) is the cost of calculating the information metric. How is the complexity of RIG compared with the complexity of the proposed method?

Author Response

We would like to thank the reviewer for taking the time to read the paper. We highly appreciate the detailed comments. We are sure that these will contribute to greatly improve the paper.

Next we respond to each of the reviewer comments. Additionally, we highlighted in blue the modified text in the paper.

The document describes how it can be distinguished among works in the literature related to information gathering and also presents the extensions to the work previously published by the authors.

This paper provides extensive explanations of the core concepts and proposed methods through detailed algorithms. It shows different evaluation metrics for each of the experiments performed. Every section is well written and in general provides plenty of analysis for each of the parts of the method.

Thank you very much for the positive comments. We highly appreciate them.

The organisation general of the paper makes it difficult to follow, mainly in sections V and VI (that appear as 5 and 6).

Thank you for this comment. In the revised version of the paper, we introduced modifications and further explanations to sections 5, 6, which we highlighted in blue. We hope that with  the introduced modifications the paper is now easy to follow.

Myopic and Random approaches are not well explained and at a first look, it results counter intuitive that a random strategy performs better than a myopic one.

We would like to thank reviewers for this comment, as it is also our goal that the paper is perfectly understood. To this end, we modified in section 6.3.2 the descriptions of myopic and random strategies to make them clear.

Additionally, we provided explaination to  why a random strategy outperforms a myopic one for the two scenarios considered. In particular, in tzhe new version of the manuscript  we wrote: “First fact that we can observe is that a random strategy outperforms a myopic one. This is due to an exploration-exploitation trade-off. A random strategy performs pure exploration:  it does not use any previously acquired data to plan next movement. This has an advantage as much more space is explored as compared to the myopic strategy. In contrast, the myopic strategy does pure exploitation as it uses the learned GP model to select the next best possible position. In this case, for the two scenarios considered, we can see that exploration is more important than exploitation. In fact, Alg. 1 balances between exploration and exploitation. This results in a higher performance of Alg. 1 respect to the other strategies.”

A main issue is found in Fig 6 results. Alg. 1 and Alg. 1* show a very similar behaviour after a few tens of seconds. Can this be interpreted as Alg. 1 has reached almost all the information about the process as Alg. 1*? This is difficult to see as there is no information of the percentage of exploration of the map. It can be assumed that a considerable proportion of the environment has been explored and Alg 1 can estimate information gain as good as Alg. 1*.

This is an important issue indeed.

First of all, we would like to comment on reviewer’s last statement “It can be assumed that a considerable proportion of the environment has been explored and Alg 1 can estimate information gain as good as Alg. 1*”. Here we disagree with the reviewer. We can look at Fig. 6 and observe that, right from the beginning, Alg 1 and Alg 1* already present a similar behavior.

Next we would like to explain in detail why Alg. 1 and Alg. 1* behave similarly. Fig. 6 demonstrates that Alg. 1, which learns hyperparameters online, is able to obtain a performance comparable to Alg.1*, which uses pre-learned hyperparameters. This tells us that Alg. 1 performs a correct exploration-exploitation trade-off. That is, in the beginning of the IG task, we would expect a robot to perform exploratory actions to “get a feeling” about the environment and about process of interest. Once this is done, the robot can use this knowledge to update the process model, and later exploit this knowledge. Essentially, this is what Alg. 1 does. Note that myopic and random are not able to perform a correct exploration-exploitation trade-off. This is reflected in the inferior performance of those methods in case when hyperparameters are unknown.

We modified Sec. 6.3.2 to remark the importance of this result.

If this is true, can the estimating of the parameters of the gaussian process be stopped in order to speed up the method?

This is of course  possible. In fact, this is a great possibility to reduce the computational complexity of the method. However, it comes at a price: if hyperparameter estimation has stoped prematurely, GP predictions will be suboptimal in this case. In the paper we decided to carry out a continuous learning of the hyperparameters to avoid converging to local minima and obtain optimal parameter estimates.

This paper describes as differences with a previous authors' work the following:

1.            We provide a more detailed description of the algorithms and the underlying methods.

2.            We include an analysis of the algorithm’s computational complexity.

3.            We present an in-depth evaluation of the algorithm with additional simulations. Specifically, we carry out a detailed analysis of the proposed RRT*-based informative path planner. Moreover, we include an additional scenario to test the whole exploration strategy described in the paper.

4.            We evaluate the online learning of the GPs hyperparameters.

5.            We propose a metric that benchmarks state-of-the-art algorithms according to their solution quality.

6.            We include an experiment on the exploration and reconstruction of a magnetic field using a ground robot equipped with a magnetic field sensor. We describe in detail the experimental setup and results.

After reading the previous work it can be found only a new simulation, consisting in adding a new scenario. This scenario is very similar to the reported before.

Thank you very much for the comment. At this point we must disagree with the reviewer, as we introduced multiple novelties with respect to our previous work. To make the differences clearer, we reorganized the contribution section.

The evaluation of the online learning of the GPs hyperparameters doesn't provides new findings about the method.

Here we must again disagree with the reviewer. The fact that Alg. 1 performs a correct learning of the hyperparameters is crucial to allow the robot taking intelligent actions, as robot’s actions are based on the GPs hyperparameters.

To make this clear, we modified section 6.3.3 and added additional explanations.

The metric proposed was already presented in the previous work.

Here we must say that the metric was first proposed in this paper, and was not in the previous version of the manuscript. This is in fact an important contribution as it allows us to analyze the algorithm performance, and provides new finding about the algorithms.

To highlight the importance of the solution quality metric, we worked on section 6.3.2 and added additional details and a further discussion.

We now really hope that this contribution is clear for the reviewer.

The new material is basically the experiment performed with the real robot.

Thank you for this comment. We agree with the reviewer that the experiment with the real robot is one of the contributions of this paper. In fact, it is a very important contribution as it demonstrates how Alg. 1 performs with a real robot.

In addition to the experiment with the real robot, we propose in this paper several additional contributions. These contributions we discussed them in previous points. Additionally, we modified section 1 to highlight the contributions.

In 6.2.2 the authors compare mutual information versus mean entropy. It is mentioned in other parts of the document that 40 simulations were performed but it is not clear if that was the case for this experiment and in that case it would be necessary to show the degree of variation.

Thanks for pointing this out. As reviewers mention, it corresponds to the average over 40 simulations. We modified the caption of Table 1 to make this explicit. Additionally, we added to Tables 1,2,3 the degree of variation, measured as a 3-sigma variation. This shows that differences are significant.

The paper states the method can be performed online, however the authors don't clarify what are the arguments and data that supports this affirmation.

Thank you very much for this comment. The online realization of the algorithm is one of the strengths of our approach. Therefore, it is crucial that we clarify it.

By online implementation, we imply that our algorithm does not require any preprocessing step, such as prior model building or estimation of hyperparamters. That is, the robot executes the algorithm as it collects data. This can be seen in Fig.1. This claim is backed by the experiment that we performed. In the experiment, the robot was able to explore the magnetic field without any prior information, by running our algorithm online.  We clarified and remarked the aforementioned points in the text in Sec. 1, 7.

It seems that the budget constrain is restricted to only one path and can not be defined in a general way as explained in [23].

This is indeed an important issue.. The role of the budget constraint is not restricted to a single path. In fact, its role is exactly the same as in [23]. Essentially, the budget constraint prunes those paths that exceed a user-defined budget. We included a clarification on Sec. 5.2.

Fig. 11 shows RMSE evolution in time using Random, Myopic and Alg. 1 strategies. The proposed strategy (Alg. 1) seems to obtain the lowest value but there is no explanation why a random strategy is better than a myopic one. These results repeat in Fig. 7 where the quality of the solution looks better for a random strategy.

This comment is related to a previous question raised by the reviewer. A random strategy outperforms a myopic one  due to an exploration-exploitation trade-off. A random strategy performs a pure exploration: it does not use previously acquired data to plan next path. In contrast, the myopic strategy performs a pure exploitation as it uses the learned GP model to select the next best possible position. The proposed Alg. 1 effectively balances between exploration and exploitation, which leads to  its higher performance.

Lines 535-538 state that the proposed path planner offers an increase over state of the art algorithms but it doesn't specify in which metric.

In the paper we refer to the increase in terms of the posterior entropy of the process after the robot measures along the planned path. We added this clarification to the text in Sec. 8.

A more illustrative example of estimated and ground truth of the process is required as the comparison between figures 9 and 10 (as mentioned in 502-505) is difficult to obtain.

Thank you very much for the useful comment. According to reviewers suggestion we generated a new figure (Fig. 9b) that permits a direct visual comparison between ground truth and estimation.

In [23] it is mentioned that the computational requirement for the RIG method is given by O(N+f(N)) where f(N) is the cost of calculating the information metric. How is the complexity of RIG compared with the complexity of the proposed method?

We would like to thank the reviewer for raising this question. According to the notation employed in RIG paper, our method would have a complexity O(N log(N) + f(N)). So, we can say that our method is slightly more complex. Nevertheless, it offers a superior performance compared to RIG algorithm.

We included the complexity of RIG in Sec. 6.3.2 for the mutual information metric.

Round 2

Reviewer 4 Report

The version 2 of the paper makes a significant improvement in clarifying several aspects of the previous version. Better explanations about the algorithms and more illustrative examples and data were provided.

There are still some minor issues:

Informative Path Planning incorporates an information-cost trade-off that maximises information and minimises cost. How this utility differs from the Best Local Ratio explained in "Searching Objects in Known Environments: Empowering Simple Heuristic Strategies"?

It is mentioned that the simulation setup used to validate Alg. 1 with synthetic and real data are stored as grids of 20 × 20 cells with a resolution of 5 and 10 centimetres. Were the 10 cms scenarios used to the simulations? Figure 3 only shows 5 cms ones.

In Figure 8 are shown three graphs with means and variances. Are them from three distinct repetitions? Moreover, It says the scale is logarithmic but the values in the axis look normal.

Some extensions that are desirable to see in the paper:

To have a comparative with discrete full coverage strategies.

To see the complete routes of the robot performed while exploring the environment.

To have graphs that show RMSE vs percentage of exploration of the environment (in terms of visited cells)

Author Response

The version 2 of the paper makes a significant improvement in clarifying several aspects of the previous version. Better explanations about the algorithms and more illustrative examples and data were provided.

We would like to thank the reviewer for taking the time to revise the paper. We highly appreciate the positive review. Next we address each of the reviewer additional comments. We highlighted in blue the modified text in the paper.

There are still some minor issues:

Informative Path Planning incorporates an information-cost trade-off that maximises information and minimises cost. How this utility differs from the Best Local Ratio explained in "Searching Objects in Known Environments: Empowering Simple Heuristic Strategies"?

We would like to thank the reviewer for suggesting this reference. The application that is considered in the suggested paper is indeed different from our target application. The suggested paper is focused on how to search objects, while our paper focuses on how to estimate a physical process of interest that takes place in an environment.

Nevertheless, although the two applications of interest are different, the two papers considered an information-distance trade-off. As we believe that the suggested paper is a relevant contribution for our work, we added a citation to the paper as we introduce our information-cost trade-off in Sec. 5.3.

It is mentioned that the simulation setup used to validate Alg. 1 with synthetic and real data are stored as grids of 20 × 20 cells with a resolution of 5 and 10 centimetres. Were the 10 cms scenarios used to the simulations? Figure 3 only shows 5 cms ones.

Thanks for pointing this out. It is in fact, unclear in the paper for which simulation we use each resolution. In particular, we use 5 cm resolution for simulations corresponding to Sec. 6.2, and 10 cm resolution for simulations corresponding to Sec. 6.3.

To make this clear, we modified the text in Sec. 6.1, and stated it explicitly.

In Figure 8 are shown three graphs with means and variances. Are them from three distinct repetitions? Moreover, It says the scale is logarithmic but the values in the axis look normal.

We would like to clarify the two aspects pointed out by the reviewer. In Fig. 8, each row corresponds to one of the hyperparameters that were learned by the robot. The three hyperparameters are estimated simultaneously by the robot. What we depict in the figures are the mean and variance, calculated over the 40 simulations runs, of the estimated hyperparameters. We really hope that this now becomes clear. We modified the text accordingly in Sec. 6.3.3.

Regarding the other point, we fully agree with reviewer’s comment. For clarification: we depict the logarithm of the values, but we do not use a logarithmic scale in the “y” axis. We decided not to use a logarithmic scale for visualization purposes, as we found out that the plots look better this way. We modified caption of Fig. 8 to avoid this misunderstanding.

Some extensions that are desirable to see in the paper:

To have a comparative with discrete full coverage strategies.

We fully understand the reviewer wish to see the comparison against full coverage strategies. In fact, we considered implementing a full coverage strategy.

However, we would like to clarify here why we decided not to include them in the paper. We did not include a full coverage strategy because the considered random, and especially myopic strategy, have been shown to provide superior results than a full coverage one for information gathering tasks. We would like to point out to references [1,2,14] to back this statement. As we are measuring the algorithm performance in the paper, we strongly believe that it is more relevant to include strategies that offer a high performance to be used as benchmarks.

To see the complete routes of the robot performed while exploring the environment.

We fully agree with the reviewer that routes calculated by the robot are an important source of information to better understand the algorithm. Because of this, we included robot’s routes in Fig. 4. Additionally, we can see robot’s route for the experiment in Fig. 10 as well as in the provided video. Let us point out that we only did not include robots routes for experiment in Sec. 6.3. This was essentially motivated by the fact that the robot start each simulation run at a random position, and the resulting plot would add little information. Moreover, we strongly believe that routes showed for the robot in the aforementioned sections is sufficient to understand our proposed algorithm.

To have graphs that show RMSE vs percentage of exploration of the environment (in terms of visited cells)

We agree with the reviewer that such a plot is a “nice to have”. However, we would like to argue that we think that it does not add much additional information and could be misleading. As we state in problem statement (Sec. 3), our goal is to reduce the RMSE between estimate and ground truth of a process as fast as possible. This is the essential reason why we evaluate our algorithm in terms of the RMSE vs time. Adding a plot that shows the RMSE vs percentage of exploration of the environment would make the paper confusing as coverage is not the goal of the paper.